# Antimycobacterial Activity of Solid Lipid Microparticles Loaded with Ursolic Acid and Oleanolic Acid: In Vitro, In Vivo, and Toxicity Assessments

**DOI:** 10.3390/microorganisms12112140

**Published:** 2024-10-25

**Authors:** Vinay Saini, Dulce Mata Espinosa, Alok Pandey, Vikas Dighe, Jorge Barrios Payán, Vithal Prasad Myneedu, Ivan Valdez Zarate, Dhanji P. Rajani, Lalit D. Anande, Rogelio Hernandez Pando, Rohit Srivastava

**Affiliations:** 1Nanobios Lab, Department of Biosciences and Bioengineering, Indian Institute of Technology Bombay, Mumbai 400076, India; drvinays14@gmail.com (V.S.); palok52@gmail.com (A.P.); 2Experimental Pathology Section, Department of Pathology, National Institute of Medical Sciences and Nutrition, Mexico City 14080, Mexico; dulce.matae@incmnsz.mx (D.M.E.); jorge.barriosp@incmnsz.mx (J.B.P.); 316244572@quimica.unam.mx (I.V.Z.); 3National Centre for Preclinical Reproductive and Genetic Toxicology, National Institute for Research in Reproductive & Child Health, ICMR, Mumbai 400012, India; dighev@nirrch.res.in; 4National Reference Laboratory (NTEP), Department of Microbiology, National Institute of TB and Respiratory Diseases, New Delhi 110030, India; vpm_myn@yahoo.co; 5Microcare Laboratory, Tuberculosis Research Centre, Surat 395003, India; microcaresurat@gmail.com; 6Former Medical Superintendent, GTB Hospital, Mumbai 400015, India; drlalitanande@gmail.com

**Keywords:** ursolic acid, oleanolic acid, solid lipid microparticles, anti-tuberculosis activity, in vivo acute toxicity, tuberculosis

## Abstract

Ursolic acid (UA) and oleanolic acid (OA) are hydrophobic triterpenoid isomers with demonstrated anti-mycobacterial (Mtb) and immune-regulatory properties, although their poor solubility limits clinical use. We report the development of solid lipid microparticles (SLMs) as delivery vehicles for UA and OA and evaluate their anti-Mtb efficacy in vitro and in vivo, as well as their acute toxicity. SLMs measured 0.7–0.89 µM in size, with complete in vitro release of OA and UA at 40 and 32 h, respectively. The minimum inhibitory concentration (MIC) of SLMs loaded with OA and UA was 40 µg/mL SLMs + 20 µg/mL OA + 20 µg/mL UA for drug-sensitive Mtb and 80 µg/mL SLMs + 40 µg/mL OA + 40 µg/mL UA for multidrug-resistant (MDR) Mtb. These SLMs showed an efficient reduction in Mtb burden in infected alveolar macrophages. In a murine model of late-stage progressive MDR-TB, aerosolized delivery of SLMs containing OA and UA via a metered-dose inhaler significantly reduced pulmonary bacterial loads and extended survival. In vivo, acute toxicity studies revealed no mortality or signs of toxicity. These findings demonstrate that SLMs are an optimal delivery system for terpenoids, providing potent in vitro and in vivo anti-TB activity with an excellent safety profile.

## 1. Introduction

Tuberculosis (TB) caused by *Mycobacterium tuberculosis* (Mtb) remains the leading cause of death from a single bacterial agent. Approximately one-fourth of the global population is latently infected with Mtb, with an estimated 10% at risk of developing active disease, particularly under conditions of immune deficiency, such as HIV infection or diabetes. In 2022, the World Health Organization estimated that 10.6 million people globally were living with active TB, resulting in 1.3 million deaths—a fact that places TB as the second leading infectious killer after COVID-19, surpassing HIV/AIDS [1]. Although effective anti-TB treatment has been available for over 60 years, it requires a regimen of four antibiotics administered for at least six months, which can lead to side effects including gastrointestinal, neurological, hematological alterations, and hepatotoxicity [2,3,4]. The prolonged treatment duration and associated adverse effects negatively affect patient adherence, reducing the effectiveness of the therapy and contributing to the emergence of drug-resistant TB [4,5,6,7,8,9]. In recent years, multidrug-resistant (MDR) TB and extensively drug-resistant (XDR) TB have rapidly emerged, posing a significant challenge due to their high lethality and the complexity and cost of treatment [10,11]. In this context, the development of both novel synthetic drugs and natural products derived from medicinal plants are promising sources of new anti-mycobacterial products that could be effective against MDR cases or help reduce the duration of conventional chemotherapy.

Natural products derived from plants have long been used as alternative treatments for a variety of diseases. A key feature of herbal medicines is the combination of multiple components with similar properties, often achieving synergistic effects. This is the case of ursolic acid (UA) and oleanolic acid (OA), two pentacyclic triterpenoid isomers that are widely found in various plants, medicinal herbs, and fruits, either as free acids or aglycones [12]. Triterpenoids have been used in the treatment of several conditions, including diabetes [13], cancer [14], and atherosclerosis [15], and have also shown potential as antiviral [16] and antimicrobial agents [17]. Our research has demonstrated that OA and UA, when used alone or in combination in a macrophage cell line, exhibit a synergistic effect against the drug-sensitive Mtb strain H37Rv and MDR clinical isolates [18]. Moreover, the combination of these compounds, administered subcutaneously in ultrapure olive oil, has shown anti-Mtb activity and an immune-stimulatory effect, enabling the elimination of the bacteria in Balb/c mice with advanced active pulmonary disease [18]. Notably, UA and OA have also been reported to possess hepatoprotective properties in mice exposed to anti-TB drugs [19]. The antituberculosis activity of OA has been reported in extracts from various plants, including the Mexican plant *Chamedora tejepilote* [20], the Peruvian plant *Clavija procera* (Theophrastaceae) [21], *Buddleja saliga* (Buddlejaceae), and *Leysera gnaphalodes* (Asteraceae), among others [7]. The minimum inhibitory concentration (MIC) of UA and OA has been determined for MDR or XDR strains as well as for drug-sensitive Mtb strains. For example, the methanolic extract of *Artemisia capillaris* containing UA showed inhibitory activity against both susceptible and resistant Mtb strains at a MIC of 12.5–25 µg/mL [22]. Other studies have found the MIC for UA against drug-sensitive strains to be 12.5–50 μg/mL [22,23], and 200 μg/mL for OA [24]. However, both UA and OA are non-polar, strongly fat-soluble compounds with low oral bioavailability, presenting a significant challenge for TB treatment. Therefore, improving the solubility and delivery of these triterpenes through nanotechnology presents an interesting opportunity. 

Solid lipid microparticles (SLM) are efficient colloidal nanoparticles that serve as versatile drug carriers, suitable for administration routes including oral, topical, ophthalmic, subcutaneous, intramuscular, and parenteral [25,26,27,28,29]. SLMs are particularly advantageous for drug delivery due to their biodegradable properties, drug encapsulation abilities, enhanced bioavailability, and biocompatibility [30]. Due to the liposolubility of UA and OA, their subcutaneous and oral administration is hindered by poor penetration, limited distribution, and insufficient availability in the lungs, the primary organ affected by TB. Thus, delivering these compounds directly to the lungs via aerosols is a promising approach for treating pulmonary infections. Our recent findings have demonstrated the efficient delivery and low toxicity of UA and OA to the lungs using either a metered-dose inhaler (MDI) or a dry powder inhaler (DPI) [31], suggesting that inhalation therapy with UA and OA could be a viable adjuvant treatment for TB. Given the promising anti-tuberculous properties of UA and OA and the advantages of SLMs as nanoparticle drug carriers, this study aimed to evaluate the antibacterial efficacy of SLMs encapsulating the terpenoids, both in vitro and in vivo.

Initially, we prepared optimum SLMs with suitable diameters and characteristics, followed by the estimation of an in vitro drug release profile. Subsequently, we assessed the anti-TB activity of these optimized SLMs using in vitro assays and evaluated their efficacy as antibiotics in vivo, employing a murine model of progressive pulmonary MDR-TB treated with SLMs encapsulating both terpenoids via MDI. Finally, we investigated the in vivo acute toxicity of the combination of UA and OA, both with and without SLM encapsulation.

## 2. Materials and Methods

### 2.1. Chemicals

OA (purity > 90%), UA (purity > 97%), and corn oil were obtained from Sigma Aldrich, St. Louis, MO, USA. Glyceryl monostearate (Mw = 358.56 g/mol) was sourced from Loba Chemie Pvt Ltd. (Mumbai, India), while Poloxamer 188, a block copolymer of ethylene oxide and propylene oxide (Kolliphor 188), was purchased from BASF, Ludwigshafen, Germany. Polycaprolactone (PCL; Mw = 14,000 g/mol) was obtained from Sigma-Aldrich (St. Louis, MO, USA) and polyvinyl alcohol (PVA; Mw = 85,000 g/mol) from SDFCL.

### 2.2. Preparation of Solid Lipid Microparticles (SLMs)

SLMs were prepared using a modified hot emulsion method, specifically high shear homogenization, as previously described with minor modifications [32]. Briefly, 200 mg of glyceryl monostearate in 5.0 mL chloroform and Kolliphor^®^ P 188 (1% *w*/*v*) was dissolved in distilled water and maintained at 90 °C to ensure a homogenous solution. The warm surfactant aqueous solution was then added to the liquefied lipid solution under high shear homogenization at 8000 rpm for 10 min. An oil-in-water (O/W) emulsion was formed by a phase-inversion process, and the resulting solution was subsequently cooled down to room temperature to yield the SLM suspension. Finally, this suspension was lyophilized to obtain water-free SLMs after centrifugation at 10,000 rpm for 15 min.

### 2.3. Characterization of SLMs

The Z-average and zeta potential of the SLMs were determined using a Malvern particle size analyzer (Model–Nano ZS, Malvern Instruments Limited, Malvern, UK). The prepared particles were diluted before measurement in Milli-Q water. All measurements were carried out in triplicate at 25 °C. 

The morphology and surface characteristics of the dried SLMs were observed by scanning electron microscopy (SEM; Jeol JSM-7600F, JEOL, Tokyo, Japan). The samples were coated with platinum in a nitrogen atmosphere and examined under an accelerating voltage of 20 kV.

### 2.4. Release Profile Studies of OA and UA from SLMs

Three separate experiments were conducted to study the release profiles of OA and UA with SLMs. The drug release study was performed using SLMs containing UA or OA with an accurately weighed sample (40 mg). The study conditions were set at a stirring speed of 100 rpm in a dissolution medium (0.1 M HCl, pH 7.0), with a total volume of 15 mL, maintained at 37 °C. At fixed time intervals, a 2.0 mL aliquot was withdrawn from each experiment for the analysis of OA and UA concentrations. The withdrawn volume was refilled with an equal volume of fresh 0.1 M PBS to maintain a constant volume at each interval. The samples were analyzed by spectrophotometer at 275 nm for OA and 272 nm for UA. 

### 2.5. In Vitro Anti-Mycobacterial Activity of SLMs with UA-OA

The anti-mycobacterial activity of the triterpenic acids was evaluated against the Mtb strain H37Rv (ATCC 27294), a reference drug-sensitive strain, and the MDR TB clinical isolate CIBIN99, which is resistant to all primary antibiotics: streptomycin, isoniazid, rifampicin, ethambutol, and pyrazinamide [33]. The microorganisms were cultured to log phase growth at 35 °C in Middlebrook 7H9 broth supplemented with 0.5% glycerol and 0.05% tyloxapol, enriched with 10% oleic acid-albumin, dextrose, and catalase (OADC, Sparks, MD, USA). The MIC was determined using a 96-well plate with 300,000 bacteria per well suspended in 100 μL of 7H9 culture medium (Middlebrook BD, Franklin, Lakes, NJ, USA), plus different concentrations of SLMs containing OA/UA, suspended in a volume of 100 µL. Plates were incubated in a humid chamber with shaking at 70 rpm at 35 °C for 7 and 10 days for the drug-susceptible and drug-resistant strains, respectively. After incubation, 40 µL of CellTiter 96^®^ (MTS) AQueous One Solution (Promega, Madison, WI, USA) was added under sterile conditions and allowed to incubate for 4 h. Absorbance was measured at 490 nm. Bacteria without treatment were used as a control. To confirm the absorbance results, bacteria from the wells were plated, and colony-forming units (CFUs) were counted after 21 days. The results represent the average of triplicates from two independent experiments.

To determine the bacillary loads in alveolar macrophages treated with SLMs containing OA and UA, the murine alveolar macrophage cell line MHS was co-cultured for 1 h with the Mtb strain H37Rv or MDR clinical isolate CIBIN99 at a multiplicity of infection (MOI) of 5:1. The cells were then washed three times with fresh RPMI 1640 medium (Invitrogen Life Technologies, Waltham, MA, USA) containing antibiotic–anti-mycotic (P4333; Sigma-Aldrich, St. Louis, MO, USA) to remove unphagocytosed bacteria. Infected macrophages were subsequently treated with either SLMs 80 µg/mL and OA + UA 40 µg/mL, or SLMs 40 µg/mL and OA + UA 20 µg/mL, dissolved in 100 µL, for the Mtb MDR clinical isolate CIBIN-99 or H37Rv, respectively, and incubated for 24 h. The cells were lysed with 0.1% SDS in 7H9 broth, and after 10 min, 20% bovine serum albumin (BSA) in 7H9 broth was added. CFUs were determined by plating 10-fold serial dilutions onto Middlebrook 7H10 agar media supplemented with OADC. CFUs were counted after 2–3 weeks of incubation at 37 °C in 5% CO_2_. The results are expressed as the mean of three independent experiments. 

To evaluate the immune stimulation of SLMs on macrophages, SLMs without payloads (plain) and SLMs loaded with UA and OA (SLMs 80 µg/mL + OA 20 µg/mL + UA 20 µg/mL) were incubated with alveolar macrophages for 24 and 48 h. The supernatants were then collected, and TNFα levels were determined using ELISA.

To evaluate the cytotoxicity of SLMs, the murine alveolar macrophage MH-S cell line was cultured in 10 cm^2^ cell culture dishes containing RPMI 1640 medium supplemented with 10% fetal bovine serum (FBS) and incubated at 37 °C in a 5% CO_2_ atmosphere until reaching 80% confluence. For the assay, 15,000 cells were seeded in each well of 96-well plates. After 24 h, different concentrations of plain SLM and SLMs containing OA/UA in a range of 240 - 80/80 μg/mL in serial 1:2 dilutions until dilution 40 - 10/10 μg/mL of SLM - OA/UA, respectively, were prepared. Cells were incubated for 48 h at 37 °C in a 5% CO_2_ atmosphere, washed with phosphate saline buffer (PBS) at pH 7.4, and suspended in 100 μL of RPMI medium supplemented with 10% FBS and 20 μL of resazurin sodium salt (Sigma-Aldrich, St. Louis, MO, USA) per well. Cell viability was assessed using a BioTek Epoch 2 microplate reader (Agilent Technologies, Santa Clara, CA, USA) with a wavelength of 600 nm. Untreated cells (without stimulus) served as the positive control (Ctrl+), representing 100% cell viability, while cells treated with 10% DMSO were the negative control (Ctrl−), representing 0% viability. The data were normalized to determine the cell survival rate.

### 2.6. In Vivo Anti-Tuberculosis Activity of SLMs with UA and OA

The MDR Mtb strain CIBIN/UMF was cultured in Middlebrook 7H9 medium (Becton Dickinson, Franklin Lakes, NJ, USA) as previously described. After one month of growth, the bacteria were harvested during the logarithmic phase. The bacilli were suspended in PBS (Sigma-Aldrich, St. Louis, MO, USA), aliquoted, and frozen at −70 °C until use. All procedures for cultivating and handling Mtb were carried out in Level 3 biosafety facilities at the Experimental Pathology Laboratory of INCMNSZ.

For the infection model, 8-week-old male BALB/c male mice weighing 22 g were used. The animals were provided by the INCMNSZ Department of Experimental Research and Animal Housing. Mice were anesthetized with sevoflurane (Sigma-Aldrich, St. Louis, MO, USA) vapor for bacterial intratracheal inoculation [34]. Briefly, each mouse was placed on a plate, the incisor teeth were secured with a rubber band, and a 22 G × 1 gauge cannula with a 125 mm blunt tip was introduced through the trachea. Each animal was inoculated with 250,000 CFUs of the MDR isolate suspended in 100 µL of PBS. Since mice are not a natural host of Mtb, a high bacterial dose is required to induce progressive disease. Infected mice were housed in ventilated boxes supplied with HEPA-filtered air and were handled under strict Level 3 Animal Biosecurity Protocols. Animal studies were approved by the Institutional Animal Ethics Committee (CICUAL-PAT 1976-19-24-2) of INCMNSZ, Mexico City, Mexico.

UA and OA encapsulated in SLMs were administered by aerial route using an MDI designed for mice [31]. One dose of SLMs containing 100 µg/mL of triterpenes (50 μg each) was given daily from day 60 post-infection until day 120, within a biosafety level 3 cabinet. The appropriate dose was selected based on the MIC determined in vitro (drug concentration effective in killing 1 × 10^6^ bacilli) by adjusting to the estimated number of bacilli in the lungs of the mice after two months of infection. The control group received the empty SLMs.

At days 30 and 60 post-treatment (90- and 120-days post-infection), groups of four mice were euthanized by exsanguination under anesthesia with sodium pentobarbital (Sigma-Aldrich, St. Louis, MO, USA) administered via intraperitoneal route at a dose of 210 mg/kg. The right lungs were immediately collected, frozen by immersion in liquid nitrogen, and stored at −70 °C until processing to determine the bacillary load via CFU counts. The lungs were homogenized, serial dilutions were prepared for each sample, and 10 µL of each dilution was plated on 7H10 medium (Becton Dickinson, Franklin Lakes, NJ, USA) enriched with OADC (Becton Dickinson, Franklin Lakes, NJ, USA). CFUs were counted after 21 days. The left lungs were perfused with 10% formaldehyde in PBS via endotracheal cannulation, fixed in the same solution for 24 h, and then sagittal sections were embedded in paraffin blocks. Sections of 4 μM thickness were mounted on glass slides, deparaffinized, and stained with hematoxylin-eosin. To determine the percentage of lung surface area affected by pneumonia, each slide from three different mice per experimental condition was photographed using a camera system that captured an image of the entire lung area, corresponding to 100% of the area. The pneumonic areas were then delimited and measured using the software analyzer Leica Application Suite v4.0 of an automated image analyzer (QWin Leica, Milton Keynes, UK), and the percentage of lung surface area affected by pneumonia was calculated. Ten infected mice were left untreated, and their survival was recorded to construct survival curves.

### 2.7. In Vivo Toxicity Assay

In-house bred, six- to eight-week-old non-pregnant female Holtzman rats (270–290 g) were used for this study. The study was approved by the Institutional Animal Ethics Committee (IAEC: 11/17) of the ICMR-National Institute for Research in Reproductive and Child Health, Mumbai, India. The animals were maintained under controlled conditions with a temperature of 23 ± 1 °C, relative humidity of 55 ± 5%, and a light/dark cycle of 14 h light and 10 h dark. A standard diet and sterile water were provided ad libitum.

An LD50 of greater than 300 mg/kg body weight (BW) for a mixture of UA and OA derived from *Bouvardia ternifolia* has been previously reported [35]. Therefore, a dose of 450 mg/kg of UA and OA was used in this study. All drug combinations were dispersed in corn oil. The animals were divided randomly into ten treatment groups (*n* = 6 per treatment group). Group I served as a vehicle control and was administered corn oil only, while group II-X received various combinations of OA, UA, and SLMs as outlined in Table 1.

All the treatments were administered orally through oral gavage. The animals were closely monitored for any adverse effects, including clinical signs of toxicity, changes in behavior patterns, or mortality. Body weight was recorded at the time of terminal sacrifice, and blood, vital organs, and reproductive organs were harvested. Heparinized blood was used for hematological analysis, and blood serum was subjected to serum clinical chemistry. Organ weights were recorded, and the organs were fixed in 10% neutral buffered formalin (NBF), embedded in paraffin, sectioned into 5 μM slices, and stained with hematoxylin-eosin for histopathological analysis.

### 2.8. Statistical Analysis

Data from the biochemical and hematological analyses, as well as the anti-mycobacterial therapeutic efficacy studies conducted in vitro and in vivo, were analyzed using GraphPad Prism 6.0. The results are expressed as mean ± SD. A one-way ANOVA test was performed to determine the significance between the control and treatment groups. Values were considered statistically significant at a level of *p* ≤ 0.05.

## 3. Results

### 3.1. Data for Particle Size, Electron Microscopy and Release Profile Studies of SLMs

The SEM of SLMs is shown in Figure 1A. SLMs exhibit a spherical shape with smooth surfaces. The particle size of SLMs was measured using dynamic light scattering (Malvern, Worcestershire, UK), with a mean particle size of 2.84 ± 0.68 μM and a Z potential of −45.7 ± 0.57 mV, while SLMs loaded with UA/OA had a mean particle size of 3.21 ± 0.82 µm. and a Z potential of −58.5 ± 1.3 mV (Figure 1B).

Drug release profile studies were performed using OA and UA loaded in SLMs. The samples were analyzed at 275 nm for OA and 272 nm for UA. Approximately 50% of both terpenoids were released within 7.0 h from the SLMs, with complete release of OA and UA estimated at 32 and 40 h, respectively (Figure 1C).

### 3.2. In Vitro Anti-Tubercular Activity of UA-OA Using SLMs

The MIC of OA + UA in SLMs against the Mtb strain H37Rv was determined to be SLMs 40 µg/mL and OA + UA 20 µg/mL (Figure 2). In comparison with the control, the bacterial load, as determined by CFUs, decreased three-fold in the presence of OA + UA in SLMs incubated with H37Rv. Regarding the MDR TB strain, the MIC value was SLMs 80 µg/mL and OA + UA 40 µg/mL of each terpenoid.

The toxicity of plain SLM and SLM bearing OA and UA was evaluated in alveolar macrophages. Plain SLM did not show toxicity in any of the evaluated concentrations (Figure 3A), while 70% survival was seen in macrophages incubated with SLM-bearing terpenoids at concentrations of 160 - 40/40, respectively, and 10% of macrophage survival was determined at concentrations of 240 - 60/60 (Figure 3B). 

To investigate whether OA and UA incorporated in SLMs could be phagocytosed and if the triterpenes could subsequently kill mycobacteria inside the macrophages, we determined bacterial growth by CFU quantification after incubating SLMs with alveolar macrophages for 24 and 48 h. As shown in Figure 3C, compared to control infected cells, macrophages infected with the drug-sensitive strain H37Rv or the MDR strain and exposed to plain SLMs showed a 20% reduction in bacterial load. A more significant reduction in bacillary loads was observed in macrophages incubated with SLMs containing OA and UA (Figure 3D).

To determine the efficiency of SLMs in activating macrophages to induce TNFα production, macrophages were incubated with SLMs, and supernatants were collected after 24 and 48 h to determine TNFα levels by ELISA. Plain SLMs induced activation of alveolar macrophages with TNFα production, and this effect was even greater when macrophages were exposed to SLMs containing OA + UA (Figure 3E).

### 3.3. In Vivo Anti-Tuberculosis Activity of UA and OA

The bactericidal activity of the SLMs loaded with UA and OA was tested in a model of progressive pulmonary TB in BALB/c mice infected with an MDR strain of *M. tuberculosis*. A dose of 100 µg/mL was administered daily via an aerosol dispenser device starting from day 60 post-infection until day 120, a period during which the disease had progressed to an advanced stage. Control-infected mice treated with plain SLMs showed progressive mortality, with all animals succumbing by 90 days post-infection. In contrast, 60% of mice treated with SLMs bearing UA and OA survived until the end of the experiment (120 days of infection, 60 days of treatment) (Figure 4A). In comparison with the control mice, those treated with SLMs bearing triterpenes showed a 50% reduction in bacillary loads after one month of treatment, with a further slight reduction observed after two months (Figure 4B). Additionally, pneumonia in treated mice was reduced by 40% after two months compared to one month prior (Figure 4C). 

### 3.4. Acute Oral Toxicity Study

In an acute oral toxicity study, female rats were exposed to OA, UA, and SLMs. No signs of toxicity or abnormal behavior were observed. While there was no effect on body weight, relative organ weights were slightly altered compared to the control group (Table 2). Specifically, liver weight decreased across all treatment groups, while kidney, heart, and spleen weights decreased in specific groups. Conversely, lung and uterus weights increased in some groups. The reduction in liver weight observed in all treatment groups is likely attributable to drug metabolism.

The study examined the impact of various treatments on serum biochemistry, lipid profile, and hematological parameters in rats, as well as the histology of their organs (Appendix A). These findings indicated alterations in serum biochemistry and lipid profile, but no pathological changes were detected in the organs.

## 4. Discussion

In the present study, SLMs were prepared using a hot emulsion method for the delivery of UA and OA. The use of SLMs enabled stable, biodegradable, and efficient particles for drug delivery. Considering that particles of <1 μm can remain suspended in the inspired and expired air reaching to the lung, while particles > 5 μm collide with the walls in the upper airways, and usually they are carried by ciliary flow to the mouth and reach primarily to the digestive tube, we developed SLN particles with a size of 1–5 μm that should have the maximum therapeutic effects.

The MIC of OA + UA in SLMs against the strain H37Rv was determined to be 40 µg/mL SLMs + 20 µg/mL OA + 20 µg/mL UA. Compared to the control, the bacterial load, as determined by CFU quantification, decreased three-fold in the presence of OA and UA in SLMs. For the MDR TB strain, the MIC value of SLMs bearing both triterpenes was found to be 80 µg/mL SLMs + 40 µg/mL OA + 40 µg/mL UA. The in vitro toxicity study using alveolar macrophages showed lower toxicity of SLMs bearing triterpenes, while in vivo acute toxicity results showed no mortality or adverse signs of toxicity compared to control animals. Thus, this study provides a comprehensive evaluation of the preparation, application, in vitro drug release, and in vivo acute toxicity of SLMs containing triterpenes, along with their efficacy as anti-tubercular compounds tested in vitro and in vivo. The results confirmed the antimycobacterial activity mediated by the combination of OA and UA and, for the first time, demonstrated that both terpenoids, when incorporated into SLMs, are also therapeutically effective and non-toxic.

It is now apparent that triterpenic acids exhibit a wide range of biological effects through multiple pathways. While the molecular mechanisms underlying the anti-mycobacterial activity of UA and OA have not yet been fully elucidated, it is proposed that both terpenoids induce significant abnormalities in the bacterial cell wall, which is characteristically lipid-rich [36,37], like their antilipidemic activity observed in eukaryotic cells [38]. Another important activity mediated by these terpenoid compounds is immune regulation. They are efficient anti-inflammatory natural products [39], although the immune responses they elicit can vary depending on triterpene concentrations and the biological conditions of the cells in different experimental systems [40]. UA and OA have been reported to stimulate the production of IFN-γ and TNF-α [41] via NF-kB transactivation in murine resting macrophages [42,43]. These cytokines are critical as protective factors in these infections because they control the activation of macrophages that phagocytize and eliminate mycobacteria [44,45,46]. Interestingly, we observed that plain SLMs can induce TNF-α production in alveolar macrophages, with an even higher level of production observed in SLMs containing UA and OA. This suggests that these SLMs possess a combination of direct antimycobacterial activity and the ability to activate immune regulatory responses, particularly macrophage activation, which is essential for controlling mycobacterial growth. This effect was also observed in our BALB/c mouse model, which is based on tracheal infection with a high mycobacterial dose to induce progressive disease. In this model, the initial phase of mycobacterial growth control is dominated by Th1 cytokines and TNF-α, followed by a later phase of progressive disease after one month of infection, characterized by lower expression of IFN-γ and TNF-α, progressive pneumonia, high bacillary load, and high mortality [34,47]. This murine TB model has been widely used to test various forms of therapy [48,49], including natural products [50,51]. Comparing the present results with previous studies using this murine model and human macrophages [18], it is clear that both terpenoids have more efficient bactericidal activity when they are administrated alone. However, UA and OA are strongly lipophilic and difficult to administrate. 

The administration of SLMs containing UA and OA via the aerial route in mice with late-stage progressive pulmonary TB caused by an MDR strain resulted in 60% survival after four months of infection, whereas all control-infected mice that received plain SLMs died within 90 days of infection. Mice treated with SLMs for one month exhibited more inflammatory lung consolidation than control mice, likely due to the pro-inflammatory activity of SLMs, which activated macrophages in vitro, inducing TNF-α production. However, after two months of treatment, lung consolidation and bacillary loads showed a trend toward reduction.

The in vitro release profile can provide essential information on the dosage form and behavior, including the release mechanism and kinetics, which is critical for the scientific development of new drug products. For complex dosage forms like microparticles, in vitro release testing is particularly significant. In our study, the drug release profile showed that SLMs did not affect the 50% release rate, but they did extend the complete release of OA and UA. The SEM micrographs revealed a non-porous surface, which is significant because drug release rates are typically faster from microparticles with higher porosity [52]. 

The targeted bio-distribution and localized delivery of drug molecules or biomolecules represent a promising strategy in the biomedical field [53]. However, direct interaction with blood and plasma may cause serious side effects on healthy organs, posing a major challenge in this area. In our study, oral administration of either plain drugs or SLMs to rats was found to be safe at the doses tested in an acute toxicity study. Currently, the clinical application of nanoparticles is limited, largely due to challenges in targeting the affected organ and controlling biodistribution [53]. Efficient biodistribution is essential for enhancing therapeutic efficacy and minimizing long-term toxicity. Direct administration of SLMs to the lungs, where TB most commonly manifests, using an aerosol MDI device is expected to improve therapeutic outcomes while limiting systemic toxicity by avoiding direct interaction between terpenoids and the bloodstream.

To understand the effect of this drug formulation on liver and kidney function, various serum biochemical parameters were evaluated. An increase in SGOT levels was observed in animals exposed to plain SLMs and UA + OA + SLM, compared to the control. Since lower levels of SGOT in the serum are generally favorable for overall health, we propose that a combination of these drugs may provide a valuable approach to disease treatments. Triterpenoids, including OA and UA, exhibit numerous biological functions and hold promise as potential therapeutic agents. The observed effects on certain parameters may be attributed to the influence of these compounds on hematopoiesis. Hematological parameters, which reflect bone marrow activity and intravascular effects, showed significant changes in female rats exposed to various combinations of UA and OA, both with and without SLMs, compared to controls. Hemoglobin concentrations and other hematological parameters were altered across all treatment groups as compared to the control. Hematopoiesis involves the proliferation and differentiation of precursor blood cells into various cell types essential for oxygen transport, host defense, repair, and other vital functions. Therefore, we hypothesize that enhanced hematopoiesis may contribute to the protective effects of these compounds.

No histopathological changes in vital organs were observed in mice exposed to a mixture of UA and OA at a 300 mg/kg dose administered subcutaneously in a subacute toxicity study [35]. The liver, kidneys, adrenal glands, brain, heart, lungs, uterus, spleen, and ovaries are the organs most susceptible to toxic responses. In this study, exposure to UA and OA at the specified doses did not result in histopathological alterations in these vital organs [35]. Therefore, although biochemical and hematological changes were observed, they were not severe enough to induce histological damage. This suggests that the dosage range selected in the present study is safe and that the aerosolized administration of the UA/OA mixture encapsulated in SLMs demonstrates effective therapeutic action against both drug-sensitive and MDR strains of *M. tuberculosis*. Ongoing studies are investigating the potential of this natural product treatment to shorten TB chemotherapy.

## 5. Conclusions

This study demonstrates that the combination of UA and OA exhibits a synergistic effect on the anti-TB activity against both drug-sensitive and MDR strains. SLMs containing UA and OA were successfully prepared using the hot emulsion method, and these microparticles were found to be suitable for both in vitro studies and in vivo administration. The morphology of the microparticles, as revealed by SEM micrographs, showed precise shape and uniform distribution, which plays an essential role in in vitro drug release. In the in vivo acute toxicity study, OA and UA, with and without SLMs, did not cause significant abnormalities or signs of toxicity. Therefore, the dose range selected in this study can be considered safe. This study represents a comprehensive approach to exploring new drug strategies for combating the rapidly emerging MDR TB strains and evaluating their long-term effects. However, the chronic effects of UA and OA, with and without SLMs, and a detailed study of their metabolism in vivo require further investigation.

## Figures and Tables

**Figure 1 microorganisms-12-02140-f001:**
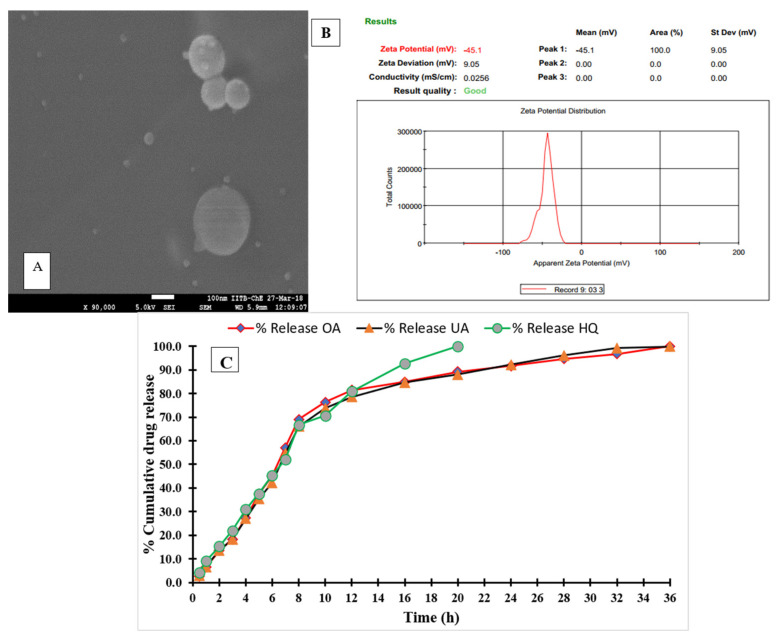
Physical characteristics of solid lipid microparticles (SLMs). (**A**) Morphology of SLMs revealed by scanning electron microscopy (SEM). (**B**) Dynamic light scattering (DLS) studies of SLMs for particle size and zeta potential measurement. (**C**) In vitro release profile of UA and OA from SLMs.

**Figure 2 microorganisms-12-02140-f002:**
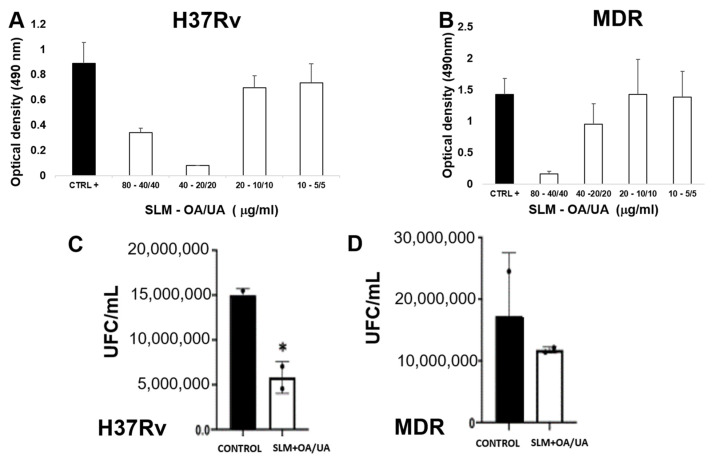
Determination of the minimum inhibitory concentration of SLMs-containing OA/UA against *M. tuberculosis*. (**A**) Absorbance of the drug-sensitive strain H37Rv at 490 nm in the absence (control) or the indicated concentrations of SLMs with triterpenes; MIC corresponds to concentration of SLMs 40 µg/mL + OA/UA 20 µg/mL. (**B**) Absorbance of the CIBIN99 MDR strain at 490 nm in the absence (control) or presence of SLMs with triterpenes; MIC was found at SLMs 80 µg/mL + OA/UA 40 µg/mL. (**C**) Colony forming units (CFUs)/mL of strain H37Rv and (**D**) MDR strain in the absence (control) or presence of SLMs containing terpenoids at the same concentration as reported in MIC determinations. The asterisk indicates statistical significance (* *p* < 0.05, two-way ANOVA).

**Figure 3 microorganisms-12-02140-f003:**
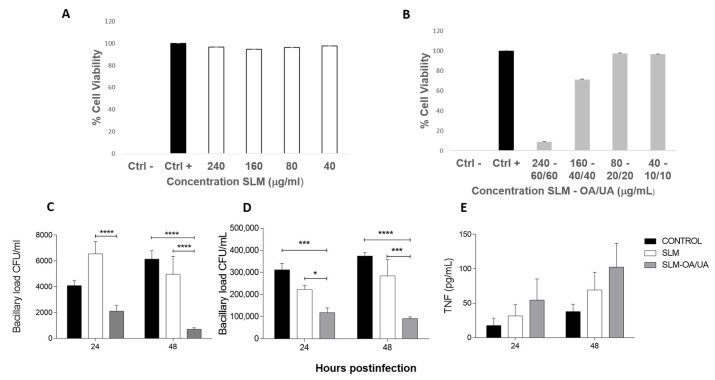
Effect of SLMs on alveolar macrophages. (**A**) Plain SLM does not have cytotoxic activity against non-infected macrophages at the indicated concentrations. (**B**) Survival of non-infected macrophages incubated with the indicated concentrations of SLM-bearing terpenoids. (**C**) Intracellular bacillary load determination by CFU quantification in macrophages infected with the drug-sensitive strain H37Rv or (**D**) MDR strain after 24 and 48 h of incubation with plain SLMs (white bars) or SLMs containing UA + OA (gray bars), compared to control macrophages (black bars). (**E**) TNFα production by non-infected macrophages incubated with SLMs, with or without OA + UA, at 24 and 48 h. Data represent the mean ± SD * *p* < 0.05, *** *p* < 0.001, **** *p* < 0.0001 were determined by two-way ANOVA Sidak’s multiple comparisons test.

**Figure 4 microorganisms-12-02140-f004:**
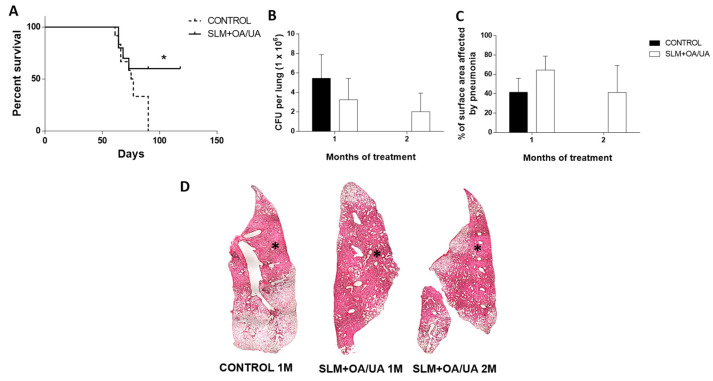
Therapeutic effect of UA and OA in SLMs in BALB/c mice following one and two months of treatment, initiated 60 days post-infection with MDR *M. tuberculosis*. (**A**) The survival rate of mice treated with SLMs containing UA and OA was significantly higher compared to control animals receiving only plain SLMs. The asterisk indicates statistical significance (* *p* < 0.05, two-way ANOVA). (**B**) Quantification of mycobacterial load by CFUs in the lungs. Treated animals showed a decrease in bacillary burden compared to the control group; there is no control group at the two-month treatment point because no animals survived to this time point. (**C**) Automated morphometry determination of the pneumonic areas that showed a slight increase in lung consolidation after one month of treatment, likely due to the pro-inflammatory effect of SLMs with triterpenes. This pulmonary inflammation decreased after two months of treatment. (**D**) Representative low-power micrographs of the indicated group; asterisks indicate areas of lung consolidation or pneumonia (40× magnification, hematoxylin/eosin staining).

**Table 1 microorganisms-12-02140-t001:** Experimental treatment groups and therapeutic doses.

Treatment Group	Drug/Compound Administered	Dose (mg/kg BW)
I	Vehicle control (corn oil)	N/A
II	Ursolic acid	450
	Oleanolic acid	450
III	Ursolic acid	450
IV	Oleanolic acid	450
V	SLMs	2000
VI	Ursolic acid	150
	Oleanolic acid	150
	SLMs	1700
VII	Ursolic acid	250
	Oleanolic acid	250
	SLMs	1500
VIII	Ursolic acid	150
IX	Oleanolic acid	150
X	Ursolic acid	150
	Oleanolic acid	150

**Table 2 microorganisms-12-02140-t002:** Effects of acute oral exposure to UA, OA, SLMs, and combination treatments on body and organ weights in female rats.

Group		Body Weight (g)	Relative Organ Weight (%)								
		Day 14	Liver	Kidney	Brain	Heart	Spleen	Lungs	Uterus	Adrenal	Ovaries
I	Control	286.5 ± 32.33	4.49 ± 0.54	0.7 ± 0.05	0.6 ± 0.07	0.44 ± 0.04	0.27 ± 0.03	0.86 ± 0.22	0.22 ± 0.06	0.01 ± 0.01	0.05 ± 0.01
II	UA + OA	286.8 ± 19.62	3.23 ± 0.27 *	0.725 ± 0.18	0.71 ± 0.13	0.46 ± 0.08	0.27 ± 0.05	2.05 ± 1.11 *	0.38 ± 0.11 *	0.04 ± 0.01	0.047 ± 0.01
V	SLM	260.67 ± 18.79	2.94 ± 0.36 *	0.71 ± 0.09	0.73 ± 0.31	0.39 ± 0.04	0.21 ± 0.03 *	0.71 ± 0.15	0.16 ± 0.02	0.02 ± 0.01	0.021 ± 0.01 *
VI	UA + OA + SLM	261.3 ± 21.49	3.36 ± 0.34 *	0.72 ± 0.07	0.71 ± 0.08	0.40 ± 0.04	0.23 ± 0.02 *	0.77 ± 0.15	0.23 ± 0.09	0.015 ± 0.01	0.03 ± 0.01

The values are represented as mean ± SD (*n* = 6); * *p* ≤ 0.05.

## Data Availability

The original contributions presented in the study are included in the article/Appendix A, further inquiries can be directed to the corresponding authors.

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
