# Peer review of "Antimycobacterial Activity of Solid Lipid Microparticles Loaded with Ursolic Acid and Oleanolic Acid: In Vitro, In Vivo, and Toxicity Assessments"

_microorganisms, 2024, doi:10.3390/microorganisms12112140_

Round 1
Reviewer 1 Report
Comments and Suggestions for Authors
In this work, the authors incorporated two plant-derived molecules (Ursolic Acid (UA) and Oleonic Acid (OA)) into lipid microparticles (SLMs). They studied the efficacy of the formulations in vitro and in vivo on antibiotic-sensitive or resistant virulent mycobacteria. There is an important need to develop new anti-tuberculosis strategies, particularly to combat antibiotic resistance. That said, several data are missing to consider the manuscript suitable for publication.
Major points:
- Results 3.1. Characterization s of SLMs: can you give the size of the different formulations tested? empty SLM and SLM with compounds at the different CCs shown in Table 1. In addition, can you comment on whether or not the size of these microparticles is advantageous for targeting the lung.
- Results 3.2. In vitro analysis: how were the MICs calculated? If you tested different concentrations, can you show all the concentrations and make a dose-response curve? What is the control in these experiments (SLMs alone)? What is the difference compared to the efficacy of unformulated OA and UA in SLMs?
- Figure 3: A: Can you comment on the results obtained at 24 hours with SLM? B: do you observe any toxicity on macrophages? C: do you have any statistical analysis?
- Figure 4A: SLM-controlled mice die after 60 days. But how do untreated mice behave? If SLMs have effects alone, it would be nice to see that they don't have a negative or positive impact on Mtb lung burden.
- Figure 4C: could you provide images of your histology ?
- The discussion et too long
Reviewer 2 Report
Comments and Suggestions for Authors
The manuscript Saini et al. describes antimycobacterial activity of ursolic and oleanolic acid. Both UA and OA are non-polar fat-soluble compounds with low oral bioavailability, so not appropriate itself for TB treatment. Therefore, the authors improved their solubility and delivery by means of drug carriers solid lipid microparticles (SLM). Optimal SLMs were prepared, loaded with UA + OA and MIC was assessed as well as acute toxicity. It was shown that UA +OA mixtureencapsulated in SLMs have effective therapeutic action against both drug-sensitive and MDR strains of M. tuberculosis. The authors suggest this natural products for shorten TB chemotherapy. UA + OA definitely has potential but as was mentioned additional investigation to evaluate chronic toxicity are needed.
The obtained results are of scientific interest and the manuscript can be published in Microorganisms after minor corrections.
Choose antimycobacterial or anti-mycobacterial and write uniformly through the text.
Line 20-21, Line 262-263, Line 158-159 should be uniformly
In the header and footer Microorganisms 2021 change for Microorganisms 2024
In the introduction according to the published data MIC of UA is 12.5-25 mg/ml and 200 mg/ml for OA/ Explain why the concentrations of UA and OA in present study are the same. Do the authors have MICs for UA and OA separately?
Round 2
Reviewer 1 Report
Comments and Suggestions for Authors
the authors addressed all my comments.
Author Response
Dear Reviewer We are submitting our revised manuscript with the minor indicated corrections in red letters.